# Conservation Humanities and Multispecies Justice

**Ursula K. Heise**

Department of English, UCLA College of Letters and Science, Division of Humanities, University of California, Los Angeles, CA 90095-1530, USA; uheise@humnet.ucla.edu

**Abstract:** This article argues that biodiversity conservation is primarily a social and cultural issue and only secondarily a scientific one. It explains the proxy logic of narratives about endangered species, which typically serve as proxies for community identities and the changes communities have undergone through processes of modernization and colonization. Polar bears, whose endangerment is interpreted differently by North American and European audiences, on the one hand, and by Inuit communities, on the other, serve as an example of how endangered species narratives not only involve culture but also, more specifically, issues of multispecies justice. Conservation humanities needs to engage with the two central problems that multispecies justice has identified and grappled with: conflicts between the interests of disadvantaged human communities and nonhuman species and conflicts and trade-offs between the interests of different nonhuman species. The essay argues that adopting the framework of "multispecies justice" rather than "conservation" will help to overcome some of the impasses of interdisciplinary collaboration in environmental studies in the past.

**Keywords:** multispecies studies; multispecies justice; conservation biology; conservation humanities; biodiversity; endangered species; biodiversity narrative; environmental humanities





## 1. The Changing Stories of Polar Bears: Conservation and Culture

Polar bears have had it. Or at least that is what it has seemed like to North American and Western European publics between the mid-1990s and the mid-2010s. Polar bears seemed not to be faring well materially: with images of polar bears forlornly stranded on melting ice flows or foraging hungrily in human trash heaps, they became prime icons of the ravages of climate change. White-furred predators in Arctic landscapes of ice and snow, they seemed perfectly adapted to their habitat until rising temperatures and diminishing sea ice in the Arctic summer made access to their preferred prey, ringed and bearded seals, more difficult. Davis Guggenheim and Al Gore featured an animated polar bear in their climate change documentary *An Inconvenient Truth* in 2006, and for two decades, polar bear mothers with their cute cubs, emaciated polar bears looking for food, and polar bears perched on icebergs with a melancholy gaze appeared again and again in the brochures, calendars, and annual reports of environmental organizations, connecting abstract statistics about rising average temperature with a gripping animal-interest story.

Polar bears seem to offer a typical example of how nonhuman species figure in conservation narratives. But they don't—not quite. For one thing, the International Union for the Conservation of Nature (IUCN), which maintains the IUCN Red List of Threatened Species, the most frequently used database for conservation efforts around the globe, classifies *Ursus maritimus* as "vulnerable"—that is, in a less serious category of endangerment than "endangered" or "critically endangered", and declares its overall population trend to be "unknown" (https://www.iucnredlist.org/species/22823/14871490, accessed on 23 December 2023). The Polar Bear Specialist group observed in its 2015 assessment that of the nineteen populations of polar bears, one had increased, six were stable, three had decreased, and that data were not sufficient to assess the population trends in the remaining nine populations (https://www.iucnredlist.org/species/22823/14871490#population, accessed on 23 December 2023). While this assessment does not indicate any optimism about polar

bears in the scientific community, it is rather less stark than public discourse about the species would lead an average citizen to believe.

The symbolic function of polar bears as highly visible victims of the "slow violence" of climate change (Mooallem 2013; Nixon 2011) also makes them somewhat less than typical of endangered species at large. Contrary to the common but erroneous idea that climate change and biodiversity loss are simply two different sides of the same ecological coin, most species to date are at risk because of habitat loss and pollution rather than rises in average temperatures. When polar bears were listed as threatened under the *Endangered Species Act* of the United States in 2008, they became the first species to be listed because of climate change, which caused considerable difficulties in the legal process, according to then-Secretary of the Interior Dirk Kempthorne: "'The polar bear poses a unique conservation challenge . . . With most [species], we can identify a localized threat, but the threat to the polar bear comes from global influences on sea ice'" (quoted in Greenemeier 2008). While the legal acknowledgment that polar bears are at risk from climate change bolsters the popular perception that climate change is the leading driver of biodiversity loss, the fact that they were the very first species in this category confirms on the contrary that in the global spectrum of species decline, they are—as yet—in a minority.

As icons of global ecological crisis, polar bears are also unusual in another sense. Public discussions about biodiversity loss and conservation typically focus on species that are associated with a particular cultural community's identity, past and present—whether that community is a nation-state, a geographic, cultural region, or Indigenous. Endangered or extinct species acquire their cultural meanings through a multistep proxy logic whereby select species assume a central role in narratives that reflect on the way in which a particular community has been changed through modernization, colonization, or a combination of both, and what the community has lost in the process. The Potawatomi philosopher Kyle Powys Whyte, for example, has made this point eloquently in his discussion of *Nmé* (lake sturgeon) and *Manoomin* (wild rice) in their significance for Anishinaabe culture; Bengal tigers and pandas are powerful proxies for national identity in India and China, respectively; and the extinct thylacine has become a touchstone for remembering a history of violence against Tasmanian Aboriginals and native species that many Australians today recall with shame and regret.[1] Polar bears do not function culturally in the same way: rather than highlighting the losses a particular cultural community has experienced, they allegorize global change and loss. At least they function in this way for large parts of the North American and Western European public, though not necessarily for other observers, as I will discuss shortly.

The proxy logic of narratives about endangered species and conservation is crucial to understanding their public resonance. Nonhuman species attract the attention of scientists and conservationists for a wide range of reasons, but when they become the object of public attention and debate, they tend to follow a typical pattern. The species that attract public attention are usually animals, not plants, fungi, or bacteria; they are typically vertebrates, while insects and mollusks rarely come into the limelight; and among vertebrates, they tend to be birds or mammals more frequently than reptiles, amphibians, or fish. These select species—often referred to as "charismatic megafauna" among conservationists—come to stand in for all animal and plant species. But species themselves, as even conservation scientists concede, are themselves proxies for the more abstract notion of biodiversity, which also includes diversity at the level of genes and ecosystems (Grenyer et al. 2006; Balvanera et al. 2014). Biodiversity tends to be culturally interpreted as a positive value in and of itself, an important dimension of what we value about nature more generally understood. In addition, the dimensions of nature that a community values are typically connected to how the community understands its own identity and its experiences of change and loss through modernization or colonization (see Figure 1; Heise 2016, pp. 19–54). In other words, public discussions and support for conservation cannot be dissociated from the cultural meanings that the species and ecosystems to be conserved have for particular communities. Or, to put it more strongly, when and where conservation becomes more than

a set of scientific research projects or an object of advocacy on the part of environmental interest groups, it is fundamentally an issue of cultural values and meaning.[2]

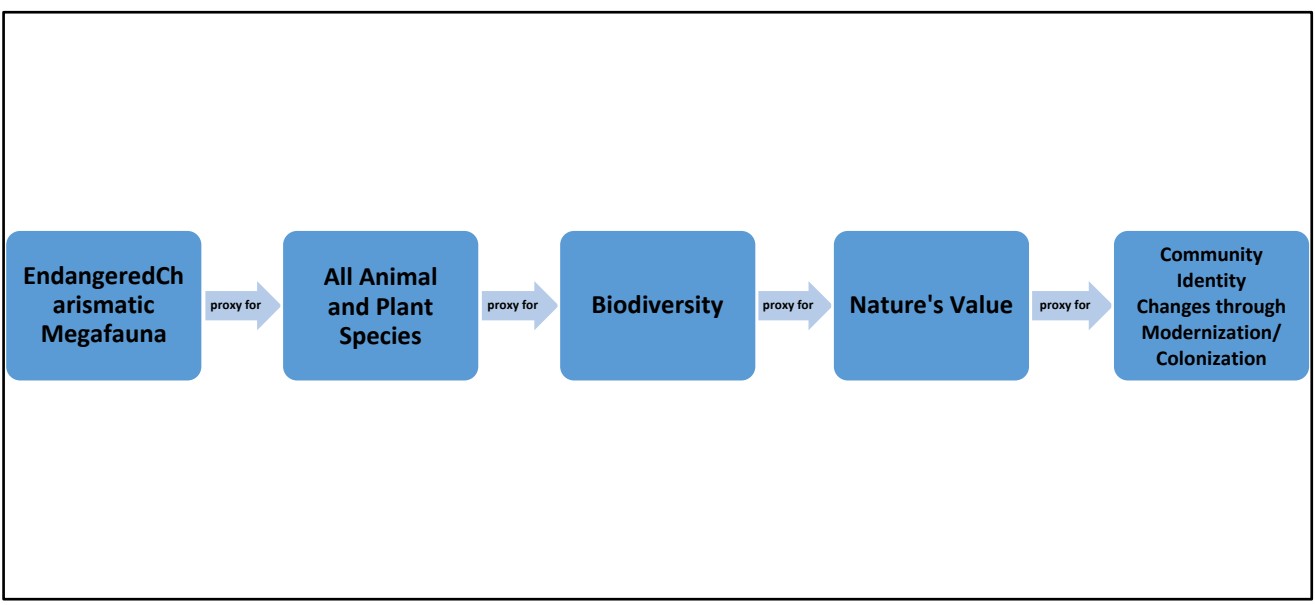

**Figure 1.** The Proxy Logic of Endangered Species Narratives.

This cultural logic of conservation is manifest in the case of the polar bears once we move beyond their perception by mainstream North American and West European media and citizens. Inuit individuals and communities with first-hand knowledge of polar bears and their habitats have been reluctant to accept the designation of the bears as threatened. The writer Roy Scranton, in an account of a cruise to the Arctic, reports that the Inuit elder Larry Audlaluk told travelers: "I want to dispel a notion about polar bears . . . I hear a lot about polar bears, a lot about what people down south think about polar bears. I want to tell you that polar bears are very healthy. There are very many polar bears, far too many of them. The ice hasn't affected our polar bears. The polar bear is just fine." (Scranton 2018, loc. 229). Scranton interprets this statement as evidence of a conflict between global environmental concerns and local economics: whereas European and North American conservationists want to preserve polar bears, Inuit are interested in preserving the income they derive from hunting bears and using or selling their pelts (Scranton 2018, loc. 255).

A much more complex narrative emerges from Zacharias Kunuk and Ian Mauro's *Qapirangajuq: Inuit Knowledge and Climate Change* (2010).[3] In this hour-long documentary featuring interviews with members of four Inuit communities in the Canadian Arctic, the camera focuses on the interviewees' accounts, mostly in Inuktitut, of their experiences of climate change. With regard to polar bears, two narratives emerge. Some interviewees assert, similarly to Audlaluk in Scranton's account, that polar bears are not endangered but on the contrary more numerous and more visible than ever before. "Those who believe the polar bear is in decline and place it on the endangered species list, they don't understand, in my opinion. Polar bears cannot be in danger. Even if at sea for a long time, they are not in danger. Because their natural environment is the sea", one interviewee says. Others indicate that they have witnessed signs of trouble: they report that polar bears have died because of overdoses of tranquilizers administered by wildlife biologists, that their hearing has been damaged by frequent helicopter noises, and that some have drowned because of the tracking collars that they were fitted with as part of the monitoring efforts. To the extent that polar bears are at risk, these interviewees assert, the cause is the intervention of "southerners" in general and wildlife biologists in particular, rather than changes in the bears' habitat.

These observations are often accompanied by expressions of contempt for biologists whose knowledge, in the interviewees' perspective, is gleaned from labs and books and ignores the lived experience of Inuit people. Noah Metuq, an interviewee from Pangnirtung, remarks with a disdainful smile that "Scientists say with great authority: 'Polar bears are in decline and will go extinct'. When I am out hunting, I never see these scientists. Not even one!" An energetic elderly woman, Rita Nashook, is even more explicit in her anger at southern intruders:

> I'm a protector of animals, a real animal activist! When animals are mistreated, I'm reminded of my late grandmother's teaching: "Unless you're going to kill an animal, do not cause it harm". Inuit are lectured: "They're endangered animals, you must not hunt them!" Inuit do not endanger animals! It's Southerners meddling with caribou, polar bears and whales that endanger wildlife! This handling and tagging is what harms animals! Wildlife biologists are the ones endangering wildlife! Then they suspect Inuit overharvesting as the cause. We are told: "You must not touch protected animals". Inuit do not endanger animals, nor do they cause needless suffering. We love our animals.

What resonates in this and a range of other comments is resentment at scientists and environmentalists who, unawares or not, repeat colonial gestures that the Inuit communities have come to know and resist for generations: the imposition of political power, authority, and knowledge from outsiders without interest in or the intent to understand and integrate local knowledge, language, and lived experience into their accounts. One may or may not agree with the interviewees' assessment that polar bears are not at risk or that the only risk they face is scientific monitoring. But it is clear that conservation, in this context, cannot succeed without awareness of the region's colonial history and respect for knowledge systems outside the natural sciences—without, in other words, the understanding that polar bears are, for Inuit communities, exactly what other endangered species are in other cultural contexts: proxies for cultural communities whose identities and ways of life have been irreversibly changed by colonization and modernization.

What this case study makes clear, in addition, is that conservation is, in many contexts, a matter of justice—in this case, the entanglement of conservation with a history of colonial oppression. I will argue in more detail shortly how the project of conservation humanities converges and conflicts with the multispecies justice framework. But shifting considerations of justice also matter for the case of the polar bear in another sense: polar bears have receded in prominence since the mid-2010s as problems of human justice have moved to the forefront of climate discourses. The geographer Mike Hulme already pointed out in 2009 that one of the discourses of climate change, which he calls "the lament for Eden", is predicated on a perception of climate as a last resort of wildness and that the loss of that wildness is construed as a diminishment of humans as well as a realm beyond them. In this context, "the polar bear—that hackneyed icon of climate change—ends up not just worrying about its own survival but is made to carry a huge additional weight on its shoulders, the weight of human nostalgia" (Hulme 2009, p. 344).

While this kind of lament has of course not disappeared, the emphasis of public discourses about climate change has shifted to questions of geopolitical justice (who causes greenhouse gas emissions and who suffers the gravest consequences of climate change in the present) and transgenerational justice (how climate change will affect those now young and those yet to be born). These forms of justice have taken on increased prominence with youth activists and movements such as Greta Thunberg, Elizabeth Wanjiru Wathuti, Extinction Rebellion, and the Green Generation Initiative, to name just a few. Similarly, how we might envision a "just transition" from current to future energy regimes that does not disadvantage the same poor communities that have borne the burden of progress in the past have moved to the forefront and, to some extent, displaced the concern over nonhuman species affected by climate change, as have the increasingly frequent and visible natural disasters that have been triggered by climate change in recent years. Polar bears as cultural icons of climate change, therefore, may be past their prime.

But that does not mean that their symbolism has stopped resonating entirely: members of Extinction Rebellion have dressed up as dead polar bears as part of their public demonstrations (Nugent 2020), and an article on young people's climate despair on the *Vice* website shows a young person sitting depressed on an ice floe, with the same iconography earlier used for polar bears (Pearl 2019)—a good example of the shift to questions of transgenerational human impacts as well as the enduring power of polar bear symbolism.[4] This does not imply that polar bears should no longer be considered in a framework of multispecies justice that includes their relations to both Inuit communities and conservationists. They should—as should other species.

## 2. Between Animals: Conservation and Justice

The questions of justice that have in recent years dominated debates about climate change in many regions have also inflected humanistic engagement with conservation. The concept of multispecies justice is rooted in a variety of intellectual strands that range from posthumanisms and new materialisms to environmental justice, multispecies ethnography, and Indigenous cosmologies.[5] Environmental justice emerged as a concept in the United States in the 1980s and was studied with a focus on environmental racism there (see Bullard [1990] 2000; Pulido 2017). Political ecology, as it was formulated in the 1990s, brought Marxian strands of thought to bear by investigating the co-production of both nature and culture by capitalist processes of production—focusing, in other words, on the causes of environmental injustice rather than the consequences which the environmental justice movement foregrounded.[6] The "environmentalism of the poor", as theorized by Ramachandra Guha and Joan Martínez-Alier in the late 1990s, made visible environmental movements in the Global South that had long combined struggles for social justice with those for environmental conservation (Guha and Martínez-Alier 1997). While some scholars continue to distinguish "environmentalism of the poor" conceptually from "environmental justice", others—myself included—use "environmental justice" to refer to movements around the globe that address environmental injustice in its various forms. Scholars such as David Pellow and David Schlosberg expanded environmental justice theory in the early twenty-first century, Schlosberg particularly by linking environmental to other theories of justice and emphasizing four basic dimensions: distributive, participatory, recognition, and capabilities justice (Schlosberg 2007).

In the early twenty-first century, anthropologists, ethologists, geographers, and philosophers in Australia, Belgium, France, Italy, and the United States moved to integrate the study of human with nonhuman forms of cognition, behavior, and sociality under such labels as "étho-ethnologie", "ethno-éthologie", or "anthropo-éthologie" (Lestel et al. 2006; Despret 2006), "transspecies theory" (Wolch et al. 1995), or "zooantropologia" (Marchesini and Tonutti 2007). The term "multispecies ethnography" was proposed in 2010 by Eben Kirksey and Stefan Helmreich as a way of foregrounding the entanglement of human societies and cultures with many nonhuman species (Kirksey 2014) and found one of its best-known applications in Anna Tsing's research on matsutake mushroom cultivation, harvesting, distribution, and consumption across cultures and continents (Tsing 2005, 2011, n.d.). The multidisciplinary study of human and more-than-human socialities is now commonly referred to as "multispecies studies" (Kirksey et al. 2016).

Multispecies justice arose in my own research on narratives about species at risk as a way of thinking about environmental justice, together with multispecies studies (Heise 2016). As Celermajer et al. have highlighted, "[a] basic democratic principle is that those affected by a policy or action should have a say in the decision-making process, but human decisions undemocratically exclude a wide range of the affected" (2021, p. 130), namely nonhumans who are not usually considered part of political communities or communities of justice. This entails

> a critical stance vis-à-vis those characteristics that are to count as criteria for moral considerability as a subject of justice. Those traditionally proffered, like agency, subjectivity, and the capacity to critically reflect and think, turn out to be—not

coincidentally—those associated with the human individual. Multispecies justice insists on the need to account for other beings with their own radically diverse life projects, capacities, phenomenologies, ways of being, functionings, forms of integrity, and relationalities . . . Multispecies justice redesigns justice away from the fiction of individualist primacy toward an ecological reality where . . . human and nonhuman animals, species, microbiomes, ecosystems, oceans, and rivers—and the relations among and across them—are all subjects of justice. (Celermajer et al. 2021, p. 127)

Envisioning what multispecies justice would look like in practice is, of course, an enormous undertaking: "to understand the types of relationships that humans ought to cultivate with more-than-human beings so as to produce just outcomes" (Celermajer et al. 2021, p. 120). This project involves the rethinking of the subject and the reinvention of political and legal procedures and institutions all the way to the acknowledgment that justice itself is a term whose definitions and practices vary greatly between cultures.

In the context of biodiversity conservation, multispecies justice engages two more concrete but no less thorny areas of research, activism, and decision-making. The first of these concerns scenarios in which the interests of particular human communities conflict with those of other species—scenarios that environmental activists and conservationists ran into head-on starting in the 1970s and 1980s. In South Asia, sub-Saharan Africa, and Latin America, for example, conservation organizations from the Global North frequently found themselves at loggerheads with local communities whose hunting and harvesting practices were impeded or prohibited when wildlife refuges, national parks, and other protected areas were created in what came to be called "fortress conservation", which in some cases led to the erosion of livelihoods and displacement (see, among many others, Agrawal and Redford 2009; Dowie 2009; Hanes 2017). In the United States, efforts to conserve the habitat of the spotted owl in the 1980s and 1990s led to extended conflicts with the timber industry and workers who perceived their jobs to be threatened for the benefit of a nonhuman species (Yaffee 1994). Today, cities across the United States contend with the opposing imperatives of building more affordable housing, creating green spaces in underprivileged neighborhoods that often lack them, and preventing sprawl that destroys natural habitats.[7] While some of these conflicts have found partial solutions, others persist unabated and will require solutions in which multispecies justice involves significant trade-offs between justice and injustice.[8]

The second major area of multispecies justice concerns in conservation involves precisely such trade-offs between different nonhuman species. Limited conservation budgets need to be allotted to some species and ecosystems at the expense of others. Native species, in many contexts, are granted greater moral consideration than non-native species, even though the non-native species often find themselves in new habitats through no agency of their own, but through human interventions. Choices sometimes need to be made between different native species: for example, when the elephant population in Kruger National Park in South Africa increased from 9000 in 1995 to 20,000 in 2008, culling was proposed as a way to limit the damage the elephants were inflicting on the vegetation, a proposal that was met with vehement resistance from animal rights activists (van Aarde et al. 1999). In a different vein, conservation initiatives themselves often inflict damage on either the endangered species themselves or on related species that are considered expendable (Van Dooren 2014; Chrulew 2017). In these and many other cases, including those where harm to a species arose unintentionally, multispecies justice thinking forces decision-makers to consider the lifeways of all involved—a challenging task even where animals of different kinds are concerned, but even more so when plants, fungi, bacteria, and other living organisms are included, as multispecies justice theorists believe they should be. As an emergent paradigm, this is the task that conservation humanities must also grapple with.

### 3. Beyond Conservation?

But are conservation humanities really an "emergent" paradigm? As even my brief overview here has shown, the humanities have long been deeply immersed in research on biodiversity and its decline. Conservation of landscapes, ecosystems, animal and plant species, and the cultures and ways of life associated with them have been concerns of environmental anthropology, history, literary studies, and philosophy for decades. Efforts to conserve individual species and fears about their extinction go back to the nineteenth century; research into mass extinctions in Earth's past in the 1970s and 1980s by paleobiologists David Raup and Jack Sepkoski led to concerns on the part of biologists such as Paul Ehrlich, Norman Myers, and E.O. Wilson that another mass extinction might currently be underway, with humans as the driving force. The coining of the term "biodiversity" in 1984 signaled how much large-scale extinction had become a focal issue for environmental movements in a variety of regions. Environmental humanists in various disciplines took up the topic to explore the definitions and values attached to biodiversity, the historically evolving cultural contexts that generate concern for particular landscapes and species, the politics and legal frameworks associated with conservation, as well as the narratives and images in which such cultural concern for the more-than-human world expresses itself. In the 2010s, with mounting concern over the loss of biodiversity as well as bioabundance, strands of research emerged that sought specifically to build on the interdisciplinarity of the environmental humanities, such as extinction studies and multispecies studies (Rose et al. 2017). Unlike other subfields of the environmental humanities, such as energy humanities or oceanic/blue humanities, the claim of conservation humanities, therefore, cannot be that it opens up previously unexplored areas of research or that its objects of research have been deprived of the scholarly attention they deserve.

Rather, the goal of conservation humanities, if the few existing publications on the paradigm are a guide, is "mainstreaming the humanities in conservation" (Holmes et al. 2021). After reviewing a selection of the published work on biodiversity in such disciplines as anthropology, cultural geography, history, literary and media studies, and philosophy, Holmes et al. observe that "these approaches have rarely been integrated into conservation science" and that "engaging with the humanities can make conservation science—-and practice—-better, and vice versa, by bringing in new questions, methods, and ways of thinking" (Holmes et al. 2021, p. 2). They concede that "[t]his may not make for comfortable or easy conversations, given that humanities research has both the capacity and the tendency to challenge key concepts, unspoken assumptions, and shibboleths in conservation" (Holmes et al. 2021, p. 2). They conclude that "[c]alls to incorporate more humanities research into conservation bodies and decision-making structures may not be new, but they need heeding more than ever" (Holmes et al. 2021, p. 4). These are conciliatory and encouraging assertions that few environmental humanists, myself included, would at first sight disagree with.

Yet I would argue that these proposals leave in place "conservation" as the major framework for thinking about the futures of species and do not engage with the reasons why humanistic and social science research on biodiversity has to date been largely ignored. Neither do they consider why a range of environmental humanists and social scientists have moved from conservation science to extinction studies, multispecies studies, or multispecies justice as framing concepts. The problem is not only that a good deal of research in environmental humanities is focused on specific communities, places, and scenarios that are difficult to generalize in ways that science finds usable nor that environmental humanists have been insufficiently concerned about linking their research to actual policy proposals—though both of these are admittedly serious obstacles. The main problem is that so long as "conservation" remains the framing term by means of which changes in biodiversity are understood, the problem remains in essence a scientific one, and scientific institutions and researchers have little reason to engage with research outside the sciences. But if, as I have argued, "biodiversity, endangered species, and extinction are primarily cultural issues, questions of what we value and what stories we tell, and only secondarily issues of science"

(Heise 2016, p. 5), the language of "mainstreaming" or "integrating" humanities approaches into the scientifically defined project of conservation is misguided. If "[a]nthropogenic biodiversity loss is, by definition, a human and social issue" (Holmes et al. 2021, p. 1), then it is the natural sciences that need to be integrated and rethought as part of the sociocultural project of multispecies justice.

So long as "conservation" remains the defining concept for the field, it will perpetuate the fundamental asymmetry that has vexed efforts at interdisciplinarity in environmental studies for decades: whereas most humanists and social scientists who research questions of biodiversity (or other issues, from soil erosion to air pollution and climate change) read a great deal of the scientific literature, most conservation scientists have never so much as opened any of the numerous books on the histories, cultures, narratives, and media that shape different communities' thinking about nonhuman species. When humanists are consulted at all in the context of interdisciplinary committees or research projects on an environmental issue, it is typically regarding the question of how to communicate scientific findings to the public: in other words, the environmental humanities are often—intentionally or unintentionally—reduced to the public relations division of the natural sciences, with little thought wasted on the idea that the humanities have research agendas, methods, and findings of their own. When Holmes et al. argue that the worlds of the environmental sciences and the environmental humanities "exist largely in parallel" without much meaningful interaction (2021, p. 2), they are right and yet gloss over a crucial difference: the two worlds are hardly symmetrical in their power to define crucial issues, their funding, or their knowledge of each other's disciplines.

Because of these foundational differences in framing, what is at issue in researching human and more-than-human relations, many—though of course not all—humanities and social-science researchers who focus on biodiversity loss prefer to approach it in terms of multispecies studies, emphasizing the role that diverse cultural communities play (and those they should play) in shaping the futures of more-than-human species and in being shaped by them, and the considerations of justice beyond the human that inform (or should inform) such futures. Some biologists and ecologists no doubt welcome such a rethinking, but many might not, because it entails that the dominant role of natural science would be diminished. Scientific findings would continue to matter, of course, in determining how ecological changes such as the decline or loss of a species affect overall ecosystem functioning, or what aspects of a habitat enable or disable a plant or animal species from flourishing in what Schlosberg calls "capabilities justice" (Schlosberg 2007, pp. 29–34). But decisions about what the ecosystems of the future should look like and what species should be helped or hindered in their survival would no longer be governed by the concept of "conservation" alone.

"Multispecies justice" not only captures more accurately what research that approaches biodiversity as a social and cultural issue needs to grapple with, it also links this research to paradigmatic shifts in other areas of environmentalism and environmental studies. Theories and activism related to environmental justice, the environmentalism of the poor, and political ecology have been successful over the last three decades in transforming environmental studies in the academy as well as environmentalism in the public sphere. From Bullard's work on toxic waste disposal siting in the 1980s and 1990s all the way to current research on, for example, climate justice, shade equity, green gentrification, and a just energy transition, considerations of environmental justice and injustice have pervasively reshaped parameters of academic study, the mission statements of environmental organizations, environmental activism, and policy. Disciplines such as public affairs, political science, economics, and law may have been more forceful drivers of this change than the humanities. But the steadily rising emphasis on concepts of justice in researching and resolving environmental crises has also revealed the power and impact of humanities research that has long engaged with environmental inequality, narratives, and images that shape ideas about justice and injustice, and creative visions of more just and sustainable multispecies societies.

Perhaps most importantly for considerations of biodiversity conservation, it has also opened a window on those strands of research in the humanities and social sciences that have most energetically questioned concepts of the human subject as separate from the natural world and other species, that have sought to critically reimagine human-animal and human-plant relations, and that have argued for an expansion of communities of justice beyond the human sphere—not least because such an expansion might also allow us to develop a better understanding of justice among humans.

Environmental narratology, the theory and analysis of fictional and nonfictional storytelling about the environment across genres and platforms that range from popular science and documentary films to novels, graphic novels, videogames, and social media, has examined a wide range of narratives that focus on multispecies societies, environmental injustice, and the forms that justice might take. Aldo Leopold, in *The Sand County Almanac*, envisioned the community of justice expanding from humans to other species in what he called the "land ethic":

> All ethics so far evolved rest upon a single premise: that the individual is a member of a community of interdependent parts . . . The land ethic simply enlarges the boundaries of the community to include soils, waters, plants, and animals, or collectively: the land . . . A land ethic of course, cannot prevent the alteration, management, and use of these "resources", but it does affirm their right to continued existence, and, at least in spots, their continued existence in a natural state.

> In short, a land ethic changes the role of *Homo sapiens* from conqueror of the land-community to plain member and citizen of it. (2001, pp. 203–4)

Many theorists and writers after Leopold (Leopold [1949] 2001) have elaborated on the complexities and contradictions that arise when what would now be called a "community of justice" is extended beyond the human. Who should be a subject of justice, on what grounds, and by whom is this determination made? Who should not be, and on what grounds? What principles should guide the resolution of contradictory claims for justice on behalf of humans as well as more-than-humans (see Section 2)?

Writers and thinkers have used a variety of metaphors and allegories, including prevalent proxy narratives, either to foreground how multispecies communities function in the present, or how they might operate in the future. Donna Haraway, after exploring such communities theoretically in her nonfiction book *Staying with the Trouble: Making Kin in the Chthulucene* (2016), veers into speculative fiction with her "Camille" series of stories, which envision genetic mixes of humans with nonhuman species as a way of figuring an ethics of sympathy and care. Sam J. Miller's *Blackfish City* (Miller 2018), set in the fictional Arctic city of Qaanaaq, features characters who are physically and psychologically bonded with orcas and polar bears via nanotechnology, signaling the emergence of new bonds between humans and animals in the face of environmental crisis that go well beyond the familiar polar-bear iconography. Sheri S. Tepper's *The Family Tree* (Tepper 1997) reverses readers' ingrained assumptions about communities of justice when people from a society 5000 years into the future time-travel to the twentieth century—at which point it becomes clear that they are not humans but the descendants of genetically engineered animals who have only experienced "ummini" (humans) as animal-like tools of labor. Starhawk's novel *The Fifth Sacred Thing* (Starhawk [Maya Greenwood] 1993) pictures an ecotopian San Francisco in the year 2048, where nonhumans are represented by humans and take part in decision-making on the city council.

Other narratives go one step further by portraying events through nonhuman voices and nonhuman points of view. Bernard Werber, a French author who has sold more than 30 million books worldwide (Desblache 2017, p. 80), has published novels that narrate events entirely or partially from nonhuman perspectives, such as those of ants and cats. In his ant trilogy, *Les fourmis* (Werber 1991), *Le jour des fourmis* (Werber 1992) and *La révolution des fourmis* (Werber 1996), which is based on meticulous research into ant cognition and

communication, he describes the world from the "point of view" of characters for whom pheromones and haptics are more important than sight or sound. Translation from ant pheromone signaling to human sound language and vice versa eventually becomes possible through technological means and enables new human-ant dialogues. Barbara Gowdy's novel *The White Bone* (1999), similarly portrays an African world of ecological change, poaching, and death entirely from the viewpoint of elephants, a world in which humans turn into "hindleggers".

Other writers reach beyond the world of animals to address the plant world, which has gained increasing importance in the environmental humanities and multispecies studies over the last decade, especially in the work of anthropologists such as Eduardo Kohn (2013) and Natasha Myers (who coined the term "Planthropocene" in 2016), philosophers such as Michael Marder (2016), Paco Calvo and Natalie Lawrence (2023), and botanists and forest managers such as Robin Wall Kimmerer (2013), Peter Wohlleben (2015), and Suzanne Simard (2021). The *grande dame* of environmental speculative fiction, Ursula K. Le Guin, had already explored *"[t]he relation of our species to plant life [which] is one of total dependence and total exploitation"* (Le Guin [1974] 1987a, p. 83; original italics) in the 1970s in stories such as "Vaster than Empires and More Slow" (Le Guin [1971] 1987b), which portrays humans' encounter with a sentient planet-wide plant community, and "Direction of the Road" (Le Guin [1974] 1987a), which features an oak tree as its narrator. More recently, Nnedi Okorafor and Tana Ford's comics series *LaGuardia: A Very Modern Story of Immigration* (Okorafor and Ford 2019) tells the story of Nigerians who migrate to the United States, with bodies composed of both human and plant genes as a metaphor for a national, racial, and cultural hybridization, but also as a meditation on the legal status of plant life.[9] Richard Powers' Pulitzer Prize-winning novel *The Overstory* (Powers 2018) explicitly translates Simard's and Wohlleben's findings on tree communities and arboreal communication into fiction by following nine human characters' encounters with different tree species as well as with protest movements against deforestation in the 1970s. Sue Burke's speculative novels *Semiosis* (Burke 2018) and *Interference* (Burke 2019) feature an intelligent bamboo that humans encounter on an alien planet. Stevland, as the bamboo comes to call itself, gradually discovers human procedures of justice and democracy in all their complexities in the midst of violent confrontations with other species. All of these narratives explore what living organisms count as members of cultural and legal communities, what moral consideration they are granted or refused on what grounds, and what concepts of justice inform ideas and initiatives that shape the lives and deaths of humans and more-than-humans.[10] In other words, they offer narrative frameworks for conservation humanities envisioned as multispecies humanities, laying the conceptual groundwork for more-than-human diplomacy, policy, and democracy.[11]

The somewhat paradoxical term "multispecies humanities" highlights one of the ironies of current interdisciplinarity in environmental studies. Ecologists and biologists today speak more often with confidence about "humans", "the human species", "human interventions", and a global "we" than environmental humanists, who tend to foreground sociocultural differences between human communities and to question the philosophical, social, and legal categorizations that separate humans out from surrounding ecologies in the first place. Multispecies justice theory has been part of this effort:

> Beyond rejecting the belief that humans alone merit ethical or political considera­tion, multispecies justice rejects three related ideas central to human exceptional­ism: (a) that humans are physically *separate or separable* from other species and nonhuman nature, (b) that humans are *unique* from all other species because they possess minds (or consciousness) and agency and (c) that humans are therefore *more important* than other species. (Celermajer et al. 2021, p. 120)

One of the contributions of the humanities to conservation is this fundamental rethink­ing of humans in relation to other species, and the accompanying emphasis on investigating and reimagining how more-than-human lives might become part of cultures and communi­ties of justice.

**Funding:** This research received no external funding.

**Data Availability Statement:** No new data were created or analyzed in this study. Data sharing is not applicable to this article.

**Conflicts of Interest:** The author declares no conflict of interest.

## Notes

1   See Whyte (2017); Schaller ([1993] 1994). For more examples of species that function as proxies for the identity of a particular cultural community, see Heise (2016, pp. 19–54).

2   In their introduction to the anthology *Extinction Studies: Stories of Time, Death, and Generations*, Rose, van Dooren, and Chrulew similarly approach extinction as "an inherently and inextricably *bicultural* phenomenon" (2017, p. 5).

3   I have discussed this documentary and the interviewees' comments on polar bears in more detail in *Imagining Extinction* (2016, pp. 241–44).

4   The illustration is by Annie Zhao.

5   Donna Haraway mentioned the term "multispecies justice" casually in *When Species Meet* (Haraway 2008) and emphasized "multispecies environmental justice" more insistently in *Staying with the Trouble* (Haraway 2016) but did not give a detailed theoretical account of these terms. For a detailed account of the intellectual sources of multispecies justice, see (Celermajer et al. 2021; Tschakert et al. 2020; Kirksey and Chao 2022).

6   I am indebted to Eric Sheppard at UCLA's Department of Geography for this distinction.

7   As an example, see the discussion of conflicting proposals for the restoration of an urban wetlands area in Los Angeles (Heise and Christensen 2020).

8   Fortress conservation, from the 1990s onward, began to be replaced by "community conservation" that seeks to ensure consultation with and benefits for local communities; whether this paradigm shift has improved biodiversity protection results continues to be a matter of debate (see Hutton et al. 2005).

9   The series *LaGuardia* was republished as a single graphic novel in 2019. For a detailed analysis, see (Sullivan 2022, pp. 348–55).

10  For a more detailed exploration of multispecies narratives, see (Heise 2016, pp. 202–36; Heise 2024); for multispecies narratives that focus on endangered and extinct species, see (Heise 2016, pp. 162–201).

11  The French environmental philosopher Baptiste Morizot has used the language of "diplomacy" as a tool for re-imagining human-wolf relationships in France after wolves remigrated into the country from Italy in the early twenty-first century (Morizot 2016; the English translation of Les diplomates is aptly titled Wildlife Diplomacy), exploring in particular ways to understand wolf thinking and to communicate with wolves.

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
