# Peer review of "Conservation Humanities and Multispecies Justice"

_humanities, doi:10.3390/h13020043_

Round 1

Reviewer 1 Report

Comments and Suggestions for Authors

This is an excellent article by an experienced scholar who reads widely in many disciplines, incorporating research from the fields of cultural geography, anthropology, ethology, philosophy, paleobiology, history, and literary and media studies. The article is clearly organized into three parts, first introducing the current state of affairs regarding conservation and the narratives that drive it, exemplified with the image and symbol of the polar bear, which is a proxy for the effects of the climate crisis. The second part starts with a useful summary of the history of the multispecies justice concept in a variety of disciplines, and the problematic current state of conservation as “fortress conservation.” The third part considers the role of the humanities for biodiversity, and for the problem of a conservation primarily driven by research in the sciences, while humanities scholarship seems to exist in a parallel world that is ignored by most scientists. The author presents a strong case for consulting and including the findings of humanists and writers of fiction and non-fiction. In order to support this argument, the author lists many examples from the works of well-known writers, from Aldo Leopold’s “land ethic” to Donna Haraway’s multispecies communities and texts by scholars and writers from the growing field of cultural and critical plant studies, as well as from fiction written from the perspective of animals and plants. The article also presents a strong conclusion as it points out the paradoxical nature of the term “multispecies humanities,” but even so considers it as “laying the conceptual groundwork” for conservation humanities and multispecies justice theory, which rejects the human/nonhuman divide prevalent in Western colonial and postcolonial societies. Though not explicitly mentioned at this point, the concept of multispecies justice is also present in many Indigenous world views (quoting Inuit voices in the first part addressing the polar bear issue is another strength of this article).

Overall this is an outstanding article, which engages with all the relevant research that has been published over time and up to the present in multiple disciplines. It presents strong arguments for a better integration of the sciences and the humanities. Because of its focus on the significance of the humanities in this scenario, the journal Humanities is the perfect place for its publication. The article is well written, accessible to scholars from many fields as well as to a general audience. It presents a clear logic of progression from the initial description of the current state of conservation to the concept of multispecies justice and why it would better serve future efforts to protect what is left of the natural world in the Anthropocene.

Author Response

I am grateful for this generous assessment of the essay.

Reviewer 2 Report

Comments and Suggestions for Authors

This paper argues in a very convincing manner  in favor of a shift of theoretical framework in environmental humanities, from a "conservation" framework primarily based on the natural sciences, to a "multi species justice" framework that takes into account symbiotic relations as well as conflicts interests between human and nonhuman species. This approach, as the author eloquently demonstrates, necessarily involves humanities research: the analysis and discussion of narrative and images that shape different groups' relation to the environment. 

The examples discussed are all very telling, and the argument is well presented, up to the point where the author discusses how land ethics thus far evolved need to change. This part, on page 9, from line 411 to line 418, needs to be further developed. It is a bit too sketchy.

The examples of novels and works of art that compels us to enlarge "the boundaries of the community" to nonhumans are very suggestive and well chosen. I was a little surprised not to find, throughout the article, any reference to the work of the French naturalist philosopher Baptiste Morizot, who calls for the development of a new sort of diplomacy with several feral species (the wolves in particular) based on a refined understanding of the way these species inhabit the earth. This is just a suggestion of a reference that could support the author's case for multi-species justice. To be added here or taken into account in future research. 

Author Response

  1. I am not sure why the reviewer calls lines 411-418 "sketchy": this is quotation from Leopold's Sand County Almanac. I believe the change in formatting from my original essay, where this long quotation was indented, has created a confusion here. But I've added a couple of sentences to explain in what ways Leopold's formulation has been superseded by more complex theorizations.
  2. I am grateful for the reminder of Morizot's work, which I have now included via a footnote and in the bibliography.